# Research Progress on Bonding Wire for Microelectronic Packaging

**DOI:** 10.3390/mi14020432

**Published:** 2023-02-11

**Authors:** Hongliang Zhou, Yingchong Zhang, Jun Cao, Chenghao Su, Chong Li, Andong Chang, Bin An

**Affiliations:** School of Mechanical and Power Engineering, Henan Polytechnic University, Jiaozuo 454000, China

**Keywords:** bonding wire, manufacturability, reliability, general comparison, development trends

## Abstract

Wire bonding is still the most popular chip interconnect technology in microelectronic packaging and will not be replaced by other interconnect methods for a long time in the future. Au bonding wire has been a mainstream semiconductor packaging material for many decades due to its unique chemical stability, reliable manufacturing, and operation properties. However, the drastic increasing price of Au bonding wire has motivated the industry to search for alternate bonding materials for use in microelectronic packaging such as Cu and Ag bonding wires. The main benefits of using Cu bonding wire over Au bonding wire are lower material cost, higher electrical and thermal conductivity that enables smaller diameter Cu bonding wire to carry identical current as an Au bonding wire without overheating, and lower reaction rates between Cu and Al that serve to improve the reliability performance in long periods of high temperature storage conditions. However, the high hardness, easy oxidation, and complex bonding process of Cu bonding wire make it not the best alternative for Au bonding wire. Therefore, Ag bonding wire as a new alternative with potential application comes to the packaging market; it has higher thermal conductivity and lower electric resistivity in comparison with Cu bonding wire, which makes it a good candidate for power electronics, and higher elastic modulus and hardness than Au bonding wire, but lower than Cu bonding wire, which makes it easier to bond. This paper begins with a brief introduction about the developing history of bonding wires. Next, manufacturability and reliability of Au, Cu, and Ag bonding wires are introduced. Furthermore, general comparisons on basic performance and applications between the three types of bonding wires are discussed. In the end, developing trends of bonding wire are provided. Hopefully, this review can be regarded as a useful complement to other reviews on wire bonding technology and applications.

## 1. Introduction

Microelectronics packaging is an essential part of the microelectronics industry, which is one of the pillar industries in countries all over the world. The common chip interconnect technologies in microelectronic packaging include wire bonding [1,2], flip-chip bonding [3,4], tape automated bonding (TAB) [5,6], etc. Wire bonding has been the most cost-effective, mature, and flexible interconnect technology in microelectronic packaging since its invention in the 1960s, which is still used to assemble more than 80% semiconductor packages in the microelectronic packaging industry [7,8]. The bonding wire, an important structural material for microelectronic packaging, plays a vital role in connecting the integrated circuit (IC) chip and the outer lead frame [9]. Statistics show that around 1/4–1/3 of the package failures are caused by wire bonding [10].

Throughout the development history of bonding wires, Au bonding wire as the earliest applied bonding wire has high mechanical strength, excellent oxidation resistance, and simple bonding process. However, owing to the steep increase in Au price and its limited performance development in recent years, the market share of Au bonding wire is decreasing yearly [11,12]. Cu bonding wire, as an alternative wire, has a potential to provide good electrical connections for high-power and highly-integrated electronics because of its higher electrical and thermal conductivity and lower cost than Au [13,14,15]. However, since the Cu bonding wire has the properties of high hardness and oxidation rate and complex bonding process that will lead to damage of pad during bond, scholars agree that Cu bonding wire is not the ultimate candidate for Au bonding wire [16,17,18]. Ag bonding wire attracts the attention of scholars because of its excellent electrical and thermal conductivity and lower price than Au. Also, Ag is one of the materials with the most potential to adapt to the development trend of microelectronic packaging with high integration, high density, and high speed, that is considered a promising bonding material for application in the packaging market [19,20]. But there still exists some reliability issues of Ag bonding wire to be solved, so it has not obtained widespread application yet [21]. This paper introduces processing techniques and reliability of the above-mentioned bonding wires, then analyzes and compares basic performance and application between the three bonding wires. Finally, we draw some conclusions and give an outlook on future challenges in microelectronic packaging, which provides technical reference for application and popularization of wire bonding in the packing field.

## 2. Au Bonding Wire

In the semiconductor manufacturing industry, Au bonding wire is the longest-serving and most widely applied bonding wire with good chemical stability, high processability, high reliability, excellent ductility, electrical conductivity, and low bonding requirements, which plays an important role in power and signal transmission. The quality of Au wire bonding is highly significant to ensure the normal operation of electronic systems [22].

### 2.1. Au Wire

The purity and diameter of Au wire have influences on its inherent performance. The higher the purity, the better the electrical conductivity and bonding ability, but the lower the elastic modulus and tensile strength, which is unfavorable for the wire loop and strength during the bonding process and will lead to bonding problems such as collapse and tailing, as shown in Figure 1 [23]. The Au wire with a varied portfolio of 2N (99%), 3N (99.9%), 4N (99.99%), etc., has a usual diameter range of 15–50 μm; the smaller the wire diameter, the more difficult the loop control. It is better to choose the purity and diameter of the Au wire according to its working performance requirements in order to ensure the bonding reliability [19]. Papadopoulos et al. [24] carried out a comparative study of 2N and 4N Au wire bond materials based on Automotive Electronics Council Qualification (AEC-Q100). The results showed that the 4N wire is reaching the lift-off values of AEC-Q100 specification at any case and is well suited up to a temperature of 150 °C, thus meeting the automotive requirements. However, the 2N wire seems to have a better performance than 4N wire due to reduced intermetallic phase (IMP) formation growth at higher temperatures.

As IC packaging develops toward miniaturization, high-density, and large-span ratio by leaps and bounds, higher demands are placed on the wire diameter, hardness, and high-temperature performance of Au wire. It is found that the problems of poor heat resistance, low recrystallization temperature, easy generation of harmful intermetallic compounds (IMCs) at high temperature and formation of Kirkendall voids during bonding process, and limited performance development, etc., of Au wire have prevented it fitting the miniaturization requirements of electronic products. Nevertheless, the above problems can be solved by doping trace elements in varied proportions into Au to make Au alloy wire [25,26].

### 2.2. Au Alloy Wire

In order to keep pace with the development of IC packaging in the direction of miniaturization, adding trace elements is an effective way to increase the strength of Au wire and reduce its wire diameter [25]. Adding Ca, Y, Be, etc., to high-purity Au can increase its recrystallization temperature, and improve its wire strength and loop stability. Adding rare earth elements can make the grain refinement, increase the size of heat-affected zone (HAZ), and improve its high temperature stability; adding Cu, Pd, Pt, etc., can inhibit the growth of IMCs and improve the bonding reliability [26]. Humpston et al. [27] developed a fine wire of the composition Au-1wt%Ti with a diameter of 25 μm through alloying, and its strength is three times higher than that of traditional Au wire. Ag and Au are infinitely miscible in solids and liquids states with a small gap between solidus and liquidus curves [28]; while a great quantity of Ag added to Au will reduce the bonding reliability between the bonding wire and the electrode of the semiconductor device, then alloying elements such as Pd, Rh, and Ru can be doped to improve bonding reliability [29]. In addition, Au alloy wire made with Au and Ni, alkaline earth metals, or rare earth metals has higher strength than traditional Au wire under the same loading conditions [30,31]. Especially, when alkaline earth metals and rare earth metals are added to Au, the strength loss of Au alloy wire is significantly reduced at high temperature [31]. Moreover, studies on the properties and applications of other Au alloy wires such as Au-Ti system, Au-Cu system, Au-Pd system, and Au-Cu-Ca system are reviewed; their advantages compared with traditional Au wire are shown in Table 1 [32,33]. Though the addition of trace elements to Au can improve bonding property of Au alloy wire, its high price due to over 50% mass fraction of Au still restricts its wide use in the global microelectronics packaging market [34,35].

### 2.3. Reliability of Au Bonding Wire

Bonding reliability is an important property of bonding wires. Compared with other types of bonding wires, Au bonding wire presents higher reliability, which is often used to connect aluminum terminals on semiconductor chips for signal transmission, and the Au-Al IMCs growth has a great impact on the bonding reliability according to many studies. Goh et al. [36] showed that the Au-Al IMCs formation makes the bonding stronger, which can improve the bonding reliability in an acidic environment. Liu et al. [37] investigated the microstructural evolution of Au-Al bonds, and the mechanical properties of Au-Al IMCs were computed by the first-principles calculations. The results showed that the Au–Al IMCs along with cracks grew during temperature increasing and aging time. Thermal exposure resulted in the transformation of the Au–Al phases associated with significant volumetric shrinkage leading to tensile stresses and promoted the growth of creep cavities that have a negative effect on the reliability of the Au–Al ball bonds [38]. In order to reduce the impact of Au-Al IMCs on the bonding reliability of electronics, scholars added trace elements such as Cu and Pd to make alloy wires to slow down the IMCs growth rate. Kim et al. [39] studied the effects on the interface reaction between Au-Pd alloy wire and Al pad, fitted the curve of IMCs thickness during thermal aging, and found that the Au-Pd alloy wire can reduce the IMCs growth rate by two orders of magnitude. Gam et al. [40] carried out a study on the growth mechanism of IMCs of Au-Pd alloy wire and Au-Cu alloy wire under high temperature storage test (HTST) at 150 °C. The results showed that both Cu and Pd can inhibit the Au-Al IMCs growth and pointed out that Cu presents a better improvement effect than Pd.

It is found that due to different atomic diffusion rates, Kirkendall voids (Figure 2 [41]) will be formed at the bonding interface, which often results in ball lifting from chip metallization and hence leads to catastrophic failure of the bond [42]. IMCs formation at the Au-Al interface caused Kirkendall voids, which resulted in interface resistance increase following non-constant Fickian law [43]. Du et al. [44] investigated the electromigration (EM) reliability and failure analysis of Au wire bonding process on Al substrate. The results showed that the interaction between the applied electric field and lattice defects increased the diffusing atomic migration and accelerated the IMCs formation. After aging at 130 °C for 200 h, Al depleted completely, and voids and cracks appeared inside the IMCs comprising Au_4_Al and Au_8_Al_3_, as shown in Figure 3. Studies on the reliability of Au wires with different diameters showed that the Kirkendall voids were predominant in finer 0.6-mil and 0.8-mil Au ball bonds. No large-sized voids were observed in 1-mil and 2-mil wires, which further were completely absent in 3-mil wire ball bonds. It indicated that finer Au wires are more likely to produce Kirkendall voids defects and may no longer satisfy the requirements for a high density and fine pitch [45].

Design and optimization of experiments (DOE) methods and finite element analysis (FEA) have been widely utilized in both industry and academia, and they are also effective and low-cost ways to study the wire bonding process and its reliability [46]. Pan et al. [47] used FEA software to simulate the Au wire bonding process and obtained the stress/strain distribution of the FAB, thus providing theoretical support for predicting the bonding strength. 3D packaging is the development trend of the future market; Chen et al. [48] proposed an effective strategy for 3D stacked packaging and verified its feasibility by FEA, which will provide useful insights for the rapid development of wire interconnection of advanced 3D stacked microelectronics packaging and the reduction of development costs. The wire loop profile is also one of the important factors that affects the bonding reliability. At present, machine learning (ML) algorithm has received great attention in the process modeling, which provides vital guidance for establishing new wire loop models [49,50,51]. Hou et al. [52] proposed and demonstrated an efficient wire loop profile prediction model based on ML algorithm and FEA. It can complete the prediction of wire profile in seconds compared to the early experimental theoretical model and FEA model prediction, providing convenience for the research of wire bonding reliability.

With the continuous development towards multi-functional and highly integrated chips, microelectronic packaging is forced to develop in the direction of fine pitch, long distance, and high performance [53,54]. In consideration of wire sweep, bond crack, wire sag, and loop instability usually happening to ultra-fine Au bonding wire used for a high density and fine pitch bonding, many researchers and scientific institutions have paid more and more attention to new alternative bonding wires of low loop, high conductivity, and ultra-fine diameter such as Cu and Ag bonding wires to be better adapted to the demands of the microelectronics packaging market [55,56].

## 3. Cu Bonding Wire

Cu bonding wire, as one of the alternatives for Au bonding wire, includes bare Cu wire, Cu alloy wire, coated Cu wire, etc. Since the beginning of the 21^st^ century, Cu-based wire has been widely used in IC packing, audio and video transmitters, active medical devices, and various electronic components; it is the key conductor material to ensure that the electrical system has steady power and signal transmission [57,58]. Compared with Au wire, Cu wire has lower cost, better electro-thermal performance, higher pull strength, and loop stability allowing for a reduced wire diameter to accommodate smaller pad sizes, and the growth rate of Cu-Al IMCs is smaller than one fifth of that of Au-Al IMCs, which greatly improves the chip frequency and bonding reliability [59,60,61].

### 3.1. Bare Cu Wire

Bare Cu wire as the conventional product of the Cu bonding wire has higher electrical conductivity, higher stiffness, and lower IMCs growing speed than Au wire, which contributes to less heat generation and void formation, better loop profile and reliability, and is more suitable for fine and ultra-fine pitch bonding [62]. It was reported that soft Cu wire slightly improves the bondability [63]. Some researchers confirmed that bare Cu wire has good reliability at an elevated temperature in high temperature storage testing [64,65] and dry atmosphere [66]. Experimental results showed that Cu bonds present higher tensile strength and shear strength than Au bonds with the same wire diameter [13,67]. Furthermore, Cu bonds potentially have higher durability than Au bonds due to the lower interdiffusion rate between the Cu and Al [68].

Although bare Cu wire has some advantages over Au wire, it still has many hurdles and cannot be used directly in industry [69]. Bare Cu wire is easy to be oxidized in air, and additional cost of forming gas, a mixture of 95% N_2_ and 5% H_2_, must be considered [70]. The higher hardness and stiffness of Cu wire over Au wire needs higher bonding force and more ultrasonic energy, which can damage the substrate, form die cratering, and induce pad peeling, pad crack, and Al splash, as shown in Figure 4 [71,72]. Wu et al. [73] showed that the critical factor leading to cracks in Cu wire bonding is that the lead frame flag floating on the thermal insert is caused by the shallower lead frame down-set or foreign matter on the thermal insert. To deal with the hurdles in Cu wire bonding, Li et al. [74] increased pad thickness to reduce the impact and rebound of the Cu bonding process and thus improved the shear strength of Cu wire bonding. A new capillary was developed to improve stitch bondability, and satisfactory results were confirmed though stitch pull and ball shear tests [70]. It was found that high-purity Cu wire can improve its basic performance and bonding reliability. However, the high purity of Cu wire will give rise to its high cost, which further limits the use of bare Cu wire in microelectronic packaging. To expand the market of Cu bonding wire, many researchers dope trace elements to the Cu matrix to make alloy wires or coat other elements on its surface to make coated Cu wires [16,75,76].

### 3.2. Cu Alloy Wire

Unlike bare Cu wire, Cu alloy wire is produced with a Cu base and trace elements such as alkali metals, alkaline earth metals, transition metals, and rare earth elements in varying concentrations that give it desired properties [77]. A patent revealed that Cu alloy wire added with Li 0.008–1.0 wt% and Ce 0.3–0.5 wt% had good corrosion resistance and short HAZ length that satisfied the requirements of high density and multi-layered packaging [78]. Fang et al. confirmed that Ca can improve the breaking resistance of Cu alloy wire and reduce its loop height. Meanwhile, Ce or Ti can improve the oxidation resistance and bondability of Cu alloy wire [79]. Yang et al. [80] drew the Cu and Cu-Fe alloy wires at room temperature, and the properties were characterized and analyzed. The results show that the strength and ductility of Cu-Fe wires are higher than those of Cu wire under the same drawing strain. Cu alloy wire with La 0.0008–0.002 wt%, Ce 0.001–0.003 wt%, Ca 0.002–0.004 wt%, and Cu 99.99–99.995 wt% had low hardness, low loop height, and good oxidation resistance that enabled it to satisfy the developing demands of high performance, multifunction, miniaturization, and portability of electronic packaging [81]. Another patent revealed that the adding of In, B, Bi, Ge, and Si to Cu matrix can increase the recrystallization temperature of Cu alloy wire and reduce its grain boundary motion rate that protected its grain boundary from cracking and reduced the damage to chip [82].

Particularly, it is well known that Cu-Ag alloys have excellent combination properties of high strength and high conductivity, and Ag has less influence on the conductivity of Cu alloys [83,84,85], which makes them the ideal materials for preparing fine bonding wires [86,87]. Continuous columnar-grained Cu-Ag alloy wires with identical diameter 0.3 mm but different Ag contents (6 wt%, 12 wt%, 24 wt%) were obtained at room temperature without intermediate annealing; the study results showed that incremental Ag contents contribute to significant improvement in tensile strength, while there is little variation in the elongation and the conductivity of the alloys, as shown in Figure 5 [88]. Xie et al. investigated the strengthening mechanisms of cold-rolled pure Cu and Cu-Ag alloys; the results verified that Ag addition increases the limiting concentration of dislocations and subgrain boundaries in the Cu solid solution, which leads to higher strength in the cold-rolled Cu-Ag alloys than that in pure Cu. Moreover, higher Ag content gives rise to higher strength of Cu-Ag alloys [89]. Zhu et al. [84] prepared a Φ40 μm Cu-4wt%Ag alloy wire with high strength and high electrical conductivity by continuously directional solidification and cold drawing. The main strengthening mechanism was grain refinement strengthening according to the calculation results of the strengthening model. The amount of trace elements added to Cu base should be controlled strictly in accordance with the bonding requirements; otherwise, it will have negative effects on the bonding properties of Cu alloy wire. Simultaneously, it was found that various degrees of cracks and fractures seem to happen to Cu alloy wires easily during the drawing process, which causes their low production efficiency and limits their applications in microelectronic packaging [90].

### 3.3. Coated Cu Wire

Coating the Cu wire with a pure metal layer which oxidizes slowly at processing temperature can extend shelf life and improve the corrosion resistance, bondability, and reliability of Cu bonding wire [91,92]. The coated pure metal layer comprises of noble metal elements such as Au, Ag, Pt, Pd and other metal elements of high corrosion resistance such as Ni, Co, Cr, and Ti. Among them, Pd shows stable performance and high corrosion resistance under long-term exposure to high temperature and high humidity conditions, and has good ductility, plasticity, and bending capacity, which adds an oxide-free and easily bondable surface for Cu wire [93,94]. Pd coating has excellent adhesion to Cu wire; round and stable FAB can typically be formed in a nitrogen atmosphere for Pd-coated Cu (PCC) wire [92,95]. PCC wire generally presents a much larger stitch bond window and more robust second bond, and shows better performance on Al pads in humidity stress and oxidation environments compared with bare Cu wire. It gains market share very fast as it can reduce chloride corrosion risk and resolve the limitation of bare Cu wire, especially on oxidation issues [96,97].

A PCC wire with Pd 1.35–8.19 wt% was invented by Zheng et al. that had long shelf life, high mechanical strength, and high oxidation resistance. It was beneficial to reduce the wire diameter and bonding pitch, and hence made it more applicable to high-density and multi-pin integrated circuit packaging [98]. Fang developed a new type of PCC wire with Ca, Mg, Al, and Sn added though the drawing process without intermediate annealing, which has preferential price, good plastic deformation capacity, and high pulling strength and reliability [99]. However, owing to the different mechanical properties of Cu and Pd, Pd coating is prone to flake off and bump during the drawing and annealing processes of PCC wire, as shown in Figure 6, which makes the Cu matrix become oxidized in the air, resulting in reliability decrease of the bonding wire [100].

### 3.4. Reliability of Cu Bonding Wire

Long-term reliability of Cu bonding wire is a major concern in replacing Au bonding wire. Cu-Al IMCs’ phase and growth behavior affected the Cu wire bond reliability [101,102]. It was found that the growth rate of Cu-Al IMCs is much lower than that of Au-Al IMCs due to the larger atomic size difference and smaller electronegativity difference between Cu and Al than that between Au and Al, which leads to bonding failure of Au wire bonding due to Kirkendall voids (Figure 7a), while thin Cu-Al IMCs micro-crack (Figure 7b) [103], the results showed that Cu wire bonding has higher reliability than Au wire bonding. [65,104,105]. Diffusion-controlled IMCs of CuAl and CuA1_2_ formed in the interface of the Cu-Al bonds after isothermal annealing by micro-X-ray diffraction [106]. Guo et al. [107] verified that CuAl_2_ phase emerged first on the Cu-Al interface, which was consistent with that in the Al-Cu system studied by Xu et al. [108]. The sequence of IMP transformations during isothermal annealing from 175 °C to 250 °C was investigated by high resolution transmission electron microscopy (HRTEM). CuAl_2_ and Cu_9_Al_4_ grew simultaneously, and the latter as a second layer (Figure 8) was the terminal product when the Al pad was completely consumed [105]. Experimental results related to interfacial structure development during the annealing process indicated that CuAl_2_ is precipitated on the Cu-Al interface initially followed by Cu_9_Al_4_, and then the reaction phases of CuAl and Cu_4_Al_3_ formed at the bond interface [109,110]. The growth mechanism of Cu-Al IMCs was investigated by Yang et al. during the annealing temperature from 50 °C to 70 °C based on the in situ HRTEM, and the results showed that Cu-Al IMCs near and apart from the Cu layer were Cu_9_Al_4_ and CuAl_2,_ respectively. An accurate growth equation of Cu-Al IMCs was also obtained according to the in situ experimental results, which provides theoretical reference for the study of Cu-Al bond reliability [92]. Moreover, the presence of Pd at the bond interface can slow down the Cu-Al IMCs growth and preserve the bond strength during thermal annealing [111].

Various papers reported that compared with Au-Al bonds, lower Cu-Al IMCs’ growth rate leads to less heat generation, lower electrical contact resistance, and better reliability and device performance [112]. Three bond pad metallizations (Al, AlCu, and AlCuSi) were used to compare Cu and Au wire bonding reliability under tests of operating life, temperature cycling, high temperature storage (HTS), thermal shock, pressure temperature humidity under bias (PTHB), temperature humidity under bias (THB), and Humidity Accelerated Stress Test (HAST), and the results showed that Cu wire bonding reliability is at least equal to the conventional Au wire bonding reliability [93]. However, the low bond strength and poor bondability at second bonds caused by surface oxidation of Cu wire limit its application in microelectronic packaging with high integration. To solve the above problems, Yuan et al. developed a Cu alloy bonding wire with Cu 99.75–99.96 wt%, W 0.01–0.1 wt%, Ag 0.01–0.03 wt%, Sc 0.01–0.02 wt%, Ti 0.001–0.03 wt%, Cr 0.001–0.03 wt%, and Fe 0.001–0.02 wt%, which has outstanding oxidation resistance and corrosion resistance, excellent plasticity and ballability, high electrical conductivity and thermal conductivity, and high strength and bonding reliability that enable it to meet the requirements of electronic packaging on high performance, multifunction, miniaturization, and low cost [113]. Murali et al. [114] added element of corrosion resistance to Cu to prepare a new alloyed Cu wire, which has homogeneous FAB formation and similar first and second bond bondability to bare Cu wire, and exhibits better electrochemical corrosion resistance and slower interfacial diffusion than that of bare Cu wire as well as higher bonding reliability than that of Au wire after HTST at 175 °C for 2000 h. Krimori et al. revealed that Cu bonding wire with oxidation-resistant metal coating Pd presents good bondability and reliability, sufficient to replace Au bonding wire (Table 2) as well as having much lower production cost than that of Au bonding wire [115,116]. In addition, electronic flame off (EFO) current has a strong influence on Pd distribution in the FAB [117], which has an effect on Cu–Al IMCs growth and thus affects bond strength and reliability [111]. Other literature also reported that the PCC wire improves bonding reliability under high temperature and high humidity and stressed environment due to the adjustment of controls for Cu-Al interdiffusion and specific IMCs formation [66,118,119].

The bonding wire used in the automotive industry is expected to change from Au wire to Cu wire in the coming years [120], and the AEC have published a new document which specifies the minimum requirements for qualification of Cu wire interconnections used in automotive electronics applications [121]. PCC wire has been widely used in large scale integration circuit (LSI) packages, but conventional PCC wire is difficult to achieve the target of automotive application. Eto et al. [122] developed a new PCC wire by adding elements in Cu core, which improved bond strength even under severe reliability test conditions. The bond pull test is the most common way to evaluate the bond strength in the wire bonding. Sun et al. [123] developed a sensor prototype for performance evaluation and applied it in a wire bonding pull test to evaluate the bond strength and provide support for wire bonding reliability research.

To sum up, Cu bonding wire has offered many benefits over the Au bonding wire such as high electrical conductivity, long wire spans without sagging, heat sinking capabilities, and low cost [124]. But the use of Cu bonding wire also has some drawbacks, mainly due to its hardness and susceptibility of Cu-Al IMCs to corrosion. Ag bonding wire, as a potential alternative wire, has better electrical and thermal conductivity than Au and Cu bonding wires as well as a relatively moderate price [125], and has been successfully bonded in die-to-die, overhanging die, DRAM stacked die [126], and multi die serial bonding in light emitting diodes (LEDs) [127].

## 4. Ag Bonding Wire

The price of Au has soared year by year; as an alternative to Au bonding wire, Cu bonding wire has been driven by its cost effectiveness, but it has its own concerns, such as sensitivity to corrosion and propensity to oxidize [128]. Since Ag has higher thermal and electrical conductivity than Au and Cu, Ag bonding wire can be chosen as an alternative material that can deliver higher reliability, lower cost, and better manufacturability than Au and Cu bonding wire [129,130,131], and Ag bonding wire can be used in Au wire bonders of the same type [132].

### 4.1. Coated Ag Wire

Like Cu, Ag is easy to become corroded in sulfurated and oxidative environments [133]. Ag wires coated with Au or Pd were studied to improve the corrosion resistance and additional properties of pure silver, and the forms of coatings include single coating and composite coating [21]. Tseng et al. [134] showed that Au-coated Ag wire (ACA) is a promising material that can enhance bonding strength and the surface performance. The cross-sectional structure of the core material of ACA studied by Kang et al. [135] was equiaxed, which had small grain size and a large number of lath-like annealing twins, as shown in Figure 9. In terms of mechanical properties, electrical properties, and bonding properties, the test results were better than those of Au wire and pure Ag wire. Tanna et al. [129] produced a Pd coated Ag bonding wire (PCS) by a new method; the diameters of FABs produced with PCS in air are found to be more consistent than those of Au wire under the same test conditions. Further, the stitch pull force, neck pull force, and ball shear force of PCS were all higher than those of Au wire. Patent [136] developed a three-layer composed of Au, Pd, and Pt from the inside to the outside coated bonding wire, whose matrix was Ag-Au-Pd alloy. It has strong oxidation and sulfidation resistance, high bonding stability, and excellent effect of inhibiting Ag^+^ migration. There are still some shortcomings to be solved about the coated Ag wire, such as uneven coating, coating flaking off, and broken wire, which will affect the performance of the bonding wire and reduce the reliability of the product. Moreover, the usage of strong acids, strong bases, and organic solvents in the electroplating process will produce harmful substances and cause environmental pollution [137,138]. At present, there are few mature coated Ag-based bonding wires used in the market.

### 4.2. Ag Alloy Wire

The elements alloyed with Ag include Au, Pd, Pt, Ru, Zn, etc. The addition of Au can improve the strength, oxidation resistance, high-temperature stability of the Ag alloy wires, and enlarge the bondability window. Fan et al. [139] found that with the increases of Au content, the HAZ of the Ag alloy wire decreases, the tension and shearing force of the FAB increase, and the best FAB morphology can be obtained when the Au content is 5%. Kuo et al. [140] showed that Ag alloy wires exhibit superior tensile strength and hardness but low conductivity and elongation as Au content increases, because relatively high Au solute atoms and low-angle grain boundary density cause strengthening and electron scattering. Therefore, adjusting the Au content in Ag alloy wires effectively can optimize its mechanical properties and conductivity. Adding Au and Pd into Ag wire increases its corrosion resistance and improves its oxidation behavior by reducing the metal surface activity [141,142]. Tsai et al. [143] evaluated Ag alloy wires with various Au and Pd contents; the results indicated that the breaking load of the various wire specimens increases with the Pd and Au contents, as shown in Figure 10. Further, it was determined that the ternary Ag-8Au-3Pd alloy wire with high strength, corrosion resistance, and reliability is an ideal substitute for the traditional Au wire, while the Ag-3Pd and Ag-4Pd are cost-friendly bonding wires for high-frequency integrated circuit devices. Cao et al. [144,145] studied the effect of heat treatment temperature on the properties and microstructure of Ag–4Pd alloy wires. The results showed that the elongation and tensile strength of bonding wire increase and decrease, respectively, as heat treatment temperature increases, as shown in Figure 11. Annealing twins appear in the bonding wire in the process of heat treatment, and the nucleation modes mainly include twinning nucleation and sub-crystalline annexation. Chuang et al. [146,147] showed that an innovative Ag-8Au-3Pd alloy wire produced by appropriate drawing and annealing processes has more annealing twins, finer grains, and higher durability against EM than traditional Ag-8Au-3Pd alloy wire, as shown in Figure 12, which possesses high thermal stability during high temperature exposure. The annealing twins in the innovative wire simultaneously increase the ductility and tensile strength with aging time, yet the electrical conductivity remains almost unchanged. Moreover, the innovative wire exhibits negligible grain growth after prolonged air storage at 600 °C for up to 180 min, whereas the grain size of traditional Au and Cu wires grow obviously under the same conditions, which means that annealing twins may have the potential to improve material performance [148].

Recently, ultrasonic ribbon bonding has received more and more attention in power electronic packaging compared with traditional wire bonding. The high current density tolerance and excellent heat dissipation of ultrasonic ribbon bonding, which features the flexibility of wire bonding and wider bonding parameter window, have driven the adoption of it to meet the requirements of high reliability packaging [149,150,151]. Based on the successful experience of replacing Au with Ag wires, Ag and Ag alloy ribbons are foreseeable options for high-power ICs. Chen et al. [151] studied the microstructure evolution of Ag-4Pd ribbon under different annealing conditions by electron backscatter diffraction (EBSD) technique, and discussed its mechanical properties, which provided a theoretical basis for the application of high-power IC modules. Chen et al. [152] further studied the influence of bonding parameters on the bonding performances of Ag-4Pd ribbon. The results showed that increasing the bonding power can promote the bonding surface, and the research has important guiding significance for providing feasible solutions for electronic packaging. 

### 4.3. Other Ag Alloy Wires

In order to develop Ag alloy wire that is more in line with production requirements, some scholars have tried adding other trace elements, such as Pt, Zn, and La, to the Ag matrix to make further progress. Pt, as noble metal with similar properties to Pd, can form a continuous solid solution with Ag, whose addition amount is generally 5 wt% [35]. Ag alloy wires containing rare earth elements have excellent anti-oxidation and mechanical properties [153,154]; the additions of Si, Zn, etc., could reduce the oxygen content of Ag bonding wire, improve its wettability, and make it easier to be drawn [34]. Cao et al. [155] added Zn 0.76 wt% to Ag-1.7Pd alloy wire; it was found that the coefficient of heat conductivity reduces by 16%, and the HAZ length decreases by 25%, as shown in Figure 13. Hsueh et al. [154] doped La into Ag to form Ag–La alloy; the results showed that adding La can improve its anti-oxidation capacity, reduce the FAB diameter, and increase the hardness of the matrix, which make it useful in the electronic packaging industry.

### 4.4. The Reliability of Ag Bonding Wire

Up to now, Ag bonding wire has not been widely used in microelectronic packaging industry because of its low bonding reliability, which is affected by FAB morphology, Ag^+^ migration, IMCs, doping element types, contents, and so on.

FAB morphology has a great influence on the reliability of Ag wire bonding; the poor FAB morphology, containing nodes and voids on FAB, can cause flaws at the bonding interface [130]. Cao et al. [156] studied the effect of Au layer thicknesses on FAB morphology of ACA, and obtained excellent FAB morphology when the Au coating thickness was 108 nm, as shown in Figure 14. They also proved that the FAB size of the Ag-10Au-3.6Pd alloy bonding wire became larger and larger with the increase of EFO time or EFO current, and an optimal FAB formed when the EFO current is 0.030A and the EFO time is 0.8 ms [157]. The experimental results provided that a good FAB morphology can be obtained in high current and short-time EFO process, which is beneficial to improve the bondability [158]. Therefore, setting appropriate bonding parameters is vital for FAB morphology, and it will directly have an effect on electronics reliability.

For decades, it has been known that Ag^+^ migration can cause short circuits and failure in microelectronic interconnects [159]. In order to inhibit Ag^+^ migration to improve bonding reliability effectively, the experimental results showed that the addition of 1.5–4.5 wt% Pd can decrease the Ag^+^ migration rate [160], and the effect is further enhanced due to the Au addition [161,162]. Cho et al. [163] confirmed that a PdO layer forms on the surface Pd-doped Ag wire, which acts as a barrier against Ag^+^ migration and diffusion to prevent Ag from dissolving in the surrounding water, as shown in Figure 15. Cai et al. [164] showed that a large number of annealing twins form in the Ag-4Pd alloy wire by adding Pd, which has positive effects on reducing the Ag^+^ migration rate and improving the reliability of the bonding wire. The research results of Guo et al. [165] and Chuang et al. [147] indicated that the Ag-8Au-3Pd alloy wire with annealing twins has significantly enhanced resistance to Ag^+^ migration, and Guo et al. supposed that a metal cap coating on the wire surface may be a better choice to prevent Ag^+^ migration. According to the above research, adding Pd to Ag alloy wires and preparing bonding wires with annealing twins can effectively inhibit the Ag^+^ migration and improve the product reliability. Recently, due to the development of power electronics and high-frequency communication ICs (such as fifth generation mobile networks and electric automotive), concerns about circuit reliability have further increased. Chen et al. [166] used EBSD to study the microstructure evolution of Ag alloy wire, providing a new perspective for the study of Ag^+^ migration.

In addition, the growth of IMCs at the bonding interface has a great influence on the bonding reliability. A certain number of IMCs can improve the bonding strength, while due to the poor electro-thermal properties of IMCs, excessive IMCs will weaken the bonding strength and conductivity, and affect the lifetime of electronic products [167]. Hsu et al. [168] showed that high current density will increase the growth rate of IMCs, which will make IMCs form more quickly at anode side than that at cathode side, and the IMCs include Ag_3_Al and Ag_2_Al at the bonding interface. In the high temperature aging test, the Al pad completely disappears with the increase of temperature and aging time, then Ag_3_Al begins to transform to Ag_2_Al, and finally IMCs are all Ag_2_Al [169]. The experimental results showed that with the increase of bias cycle, the growth of Ag_2_Al at the bonding interface is responsible for the increase in the resistance of the bonding wire [170]. Chuang et al. [171] proved that even after prolonged high HTS at 150 °C for 500 h, IMCs thickness of Ag-8Au-3Pd is only around 1.7 µm, which has much higher reliability than that of the Au-Al IMCs. Long et al. [172] designed and created different surface textures on glass substrates. The results showed that the oxide removal efficiency is enhanced, the IMC growth is improved, and the bonding strength is several times higher than that on smooth surfaces. The above studies show that the quantities and types of IMCs are also the keys for the reliability. Up to now, most studies focus on Ag alloy wires, some of which have been used in LED and IC packages such as Ag-8Au-3Pd and Ag-4Pd wires [143]. However, uneven mixed or unreasonable chosen alloy components will lead to many defects during application, such as sharp FAB, golf-clubbed ball, and wave ball, which have negative effects on bonding reliability. Thus, it is necessary to make great efforts to carry on further study on the reliability of Ag bonding wire.

## 5. Comparisons

As reviewed in the aforementioned sections, Au bonding wire shows the best comprehensive performance, but has the highest price. Compared with Au bonding wire, Cu and Ag bonding wires, as potential alternative wires, have better cost effectiveness and higher electrical and thermal conductivity, which enable them to satisfy the requirements for a high density and fine pitch of microelectronic packaging and gain market share fast. The performance comparison of the three types of bonding wires is shown in Table 3. All the three types of bonding wires still face some technical challenges, and numerous researchers make great efforts to solve them as shown in Table 4.

## 6. Summaries

Wire bonding is a promising and critical interconnection technology with low cost, large-span ratio, and high-density packaging capability, which has been applied in microelectronic packaging for more than 70 years. Due to the rapid bonding process, the evolution of bonding interface is difficult to observe, and the bonding mechanism cannot be clearly described. Au bonding wire is the earliest and most widely used bonding wire, but due to the limit of electro-thermal performance and high price, its market share is gradually decreasing; currently, it is mainly used in high-end products applied in aerospace and other fields. The application of Cu bonding wire came out in the 1990s, and its market share has increased rapidly in recent years, especially Pd-coated Cu wire, which has become one of the most used bonding wires in low-pin-count semiconductor packaging, Radio Frequency (RF) device packaging, micro-electromechanical systems, and so on, but some reliability problems still need to be solved. Ag bonding wire began to appear in the market in the 2010s, and it has been widely used in IC packaging and LED packaging for many years. Because of its excellent usability, scholars believe that it will replace Au bonding wire applied in high-end products. In addition, multilayer wire bonding is the development trend of the future market, and the design of wire loop profile is also the focus of research.

In general, the research tasks of bonding wires in the future may focus on the following aspects:

(1) It is difficult to observe the evolution of the bonding interface during the wire bonding process, and the wire bonding mechanism is still unclear. Therefore, developing new research techniques can play important roles in studies on the wire bonding mechanism.

(2) The electroplating and electroless plating processes currently used in the preparation of coating bonding wires will produce harmful substances that may pollute the environment, so it is necessary to develop new environmentally friendly and resource-efficient surface treatment technology to prepare coating bonding wires.

(3) Microalloying can improve the mechanical properties of bonding wires, so appropriate multi-component microalloy doping process is of significance in the preparation of bonding wires, which is beneficial to increase the market share of Cu-based and Ag-based bonding wires.

(4) The bonding wires with high cost and poor performance have not been enough to satisfy the requirements of electronic packaging on high performance, multifunction, miniaturization, and low cost, so developing Cu-based or Ag-based bonding wires with cost effectiveness and excellent performance has become more and more popular for scholars and manufacturers.

(5) The wire loop profile has a great impact on the bonding reliability. The main challenge is to control the wire loop height in the ultra-thin packaging space. It is the future trend to study the wire profile models and the reliability of wire bonding by FEA and ML algorithms.

## Figures and Tables

**Figure 1 micromachines-14-00432-f001:**
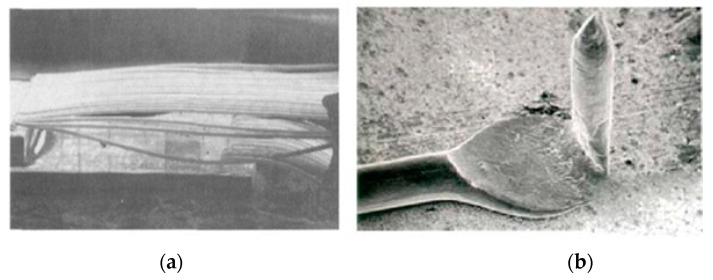
The phenomenon of wire (**a**) collapse and (**b**) tailing of Au bonding wire [23]. Reproduced with permission from Ref. [23]; published by Precious Metals, 2017.

**Figure 2 micromachines-14-00432-f002:**
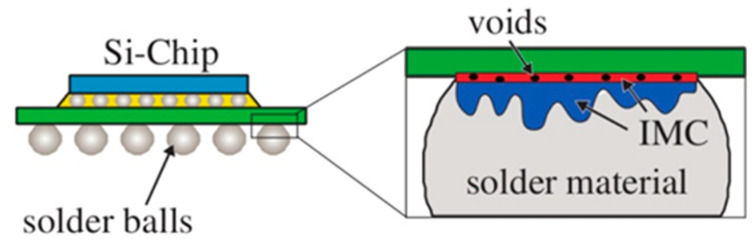
An example of Kirkendall voids in microelectronic packaging [41]. Copyright, 2020, Elsevier.

**Figure 3 micromachines-14-00432-f003:**
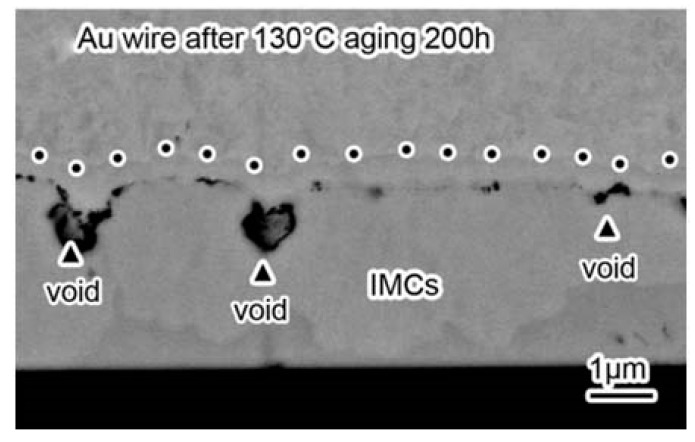
The cross-section images of bonding interface of Au wire after aging at 130 °C for 200 h [44]. Copyright, 2020, Springer Nature.

**Figure 4 micromachines-14-00432-f004:**
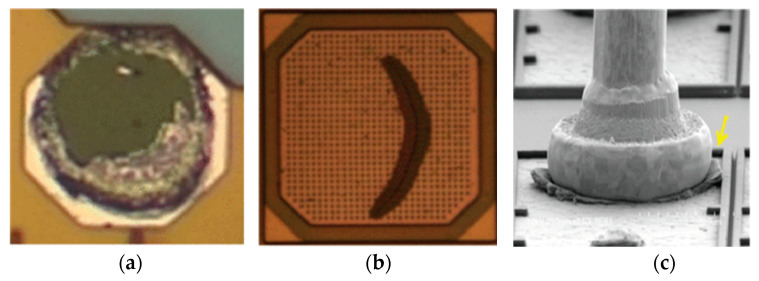
Schematic diagram of pad defects during bare Cu wire bonding: (**a**) pad peeling; (**b**) pad crack; (**c**) Al splash [72]. Copyright, 2015, IEEE.

**Figure 5 micromachines-14-00432-f005:**
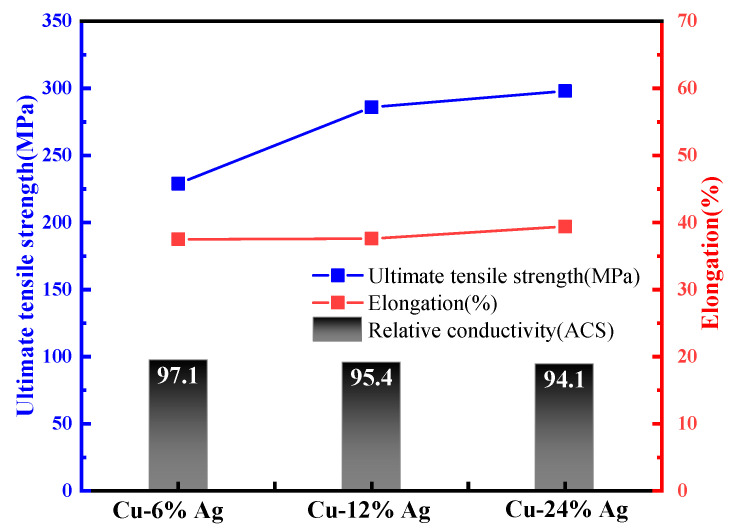
Mechanical-electrical properties of directionally solidified Cu-Ag alloys [88]. Copyright, 2018, Elsevier.

**Figure 6 micromachines-14-00432-f006:**
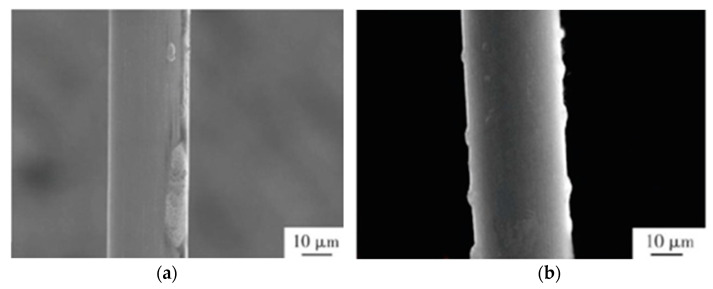
Surface morphology of Pd-coated Cu wire: (**a**) the coating flakes off; (**b**) the coating bumps [100]. Reproduced with permission from Ref. [100]; published by Heat Treatment of Metals, 2016.

**Figure 7 micromachines-14-00432-f007:**
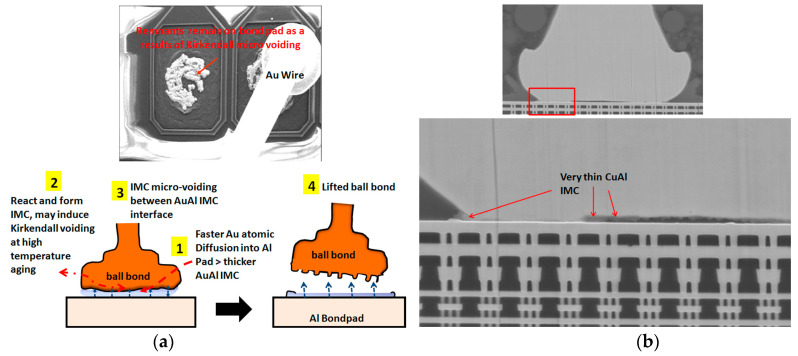
(**a**) Proposed failure mechanism of AuAl Kirkendall micro-voiding and caused lifted ball bond (**b**) SEM images show very thin CuAl IMC formation on bonded stage of Cu wirebond package prior to reliability stress. No microcracking beneath PdCu ball bond. [103]. Copyright, 2013, ASME Digital Collection.

**Figure 8 micromachines-14-00432-f008:**
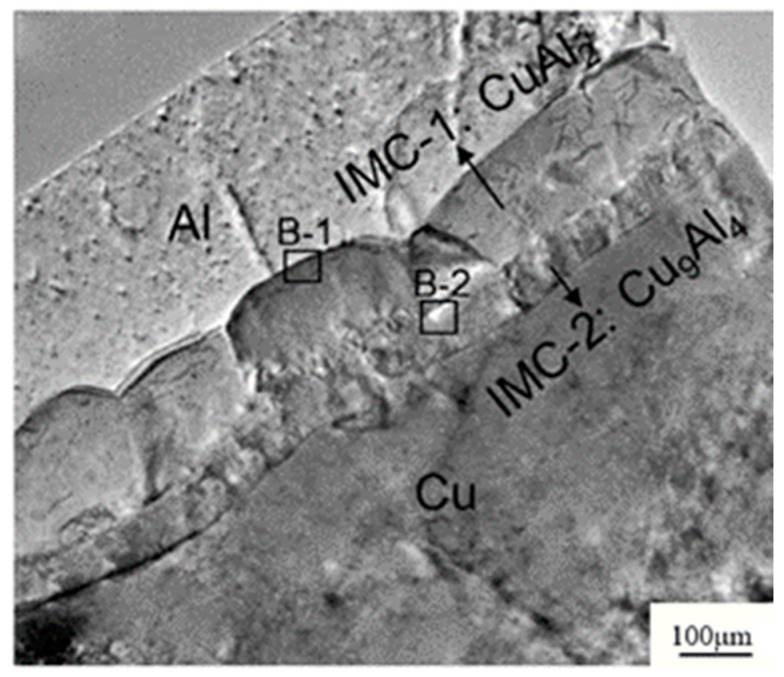
A typical TEM image of the Cu-Al interface after annealing at 175 °C for 25 h [105]. Copyright, 2011, Elsevier.

**Figure 9 micromachines-14-00432-f009:**
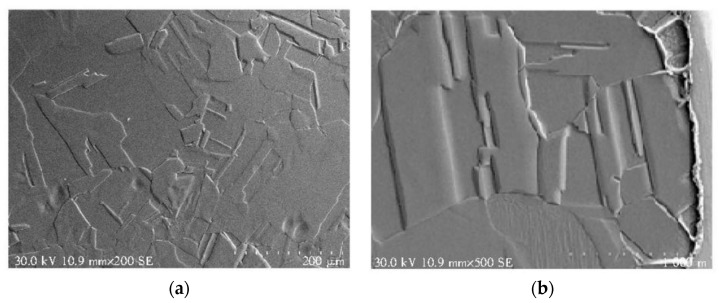
SEM image of cross section of Au-coated Ag wire, (**a**) the magnification is 200 times and (**b**) the magnification is 500 times [135]. Reproduced with permission from Ref. [135]; published by Semiconductor Technology, 2018.

**Figure 10 micromachines-14-00432-f010:**
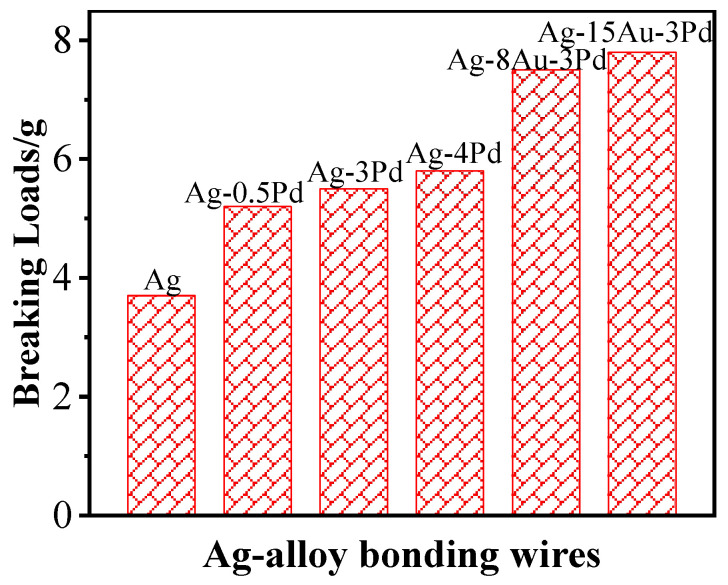
Fracture load of various Ag alloy wires [143]. Copyright, 2016, IEEE.

**Figure 11 micromachines-14-00432-f011:**
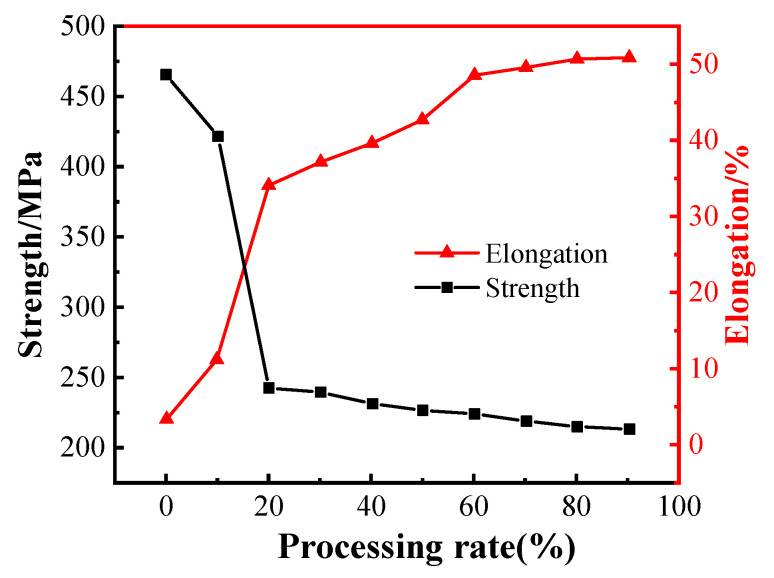
Variation curve of tensile strength and elongation of Ag-4Pd wire during cold deformation [145]. Reproduced with permission from Ref. [145]; published by Journal of Mechanical Engineering, 2016.

**Figure 12 micromachines-14-00432-f012:**
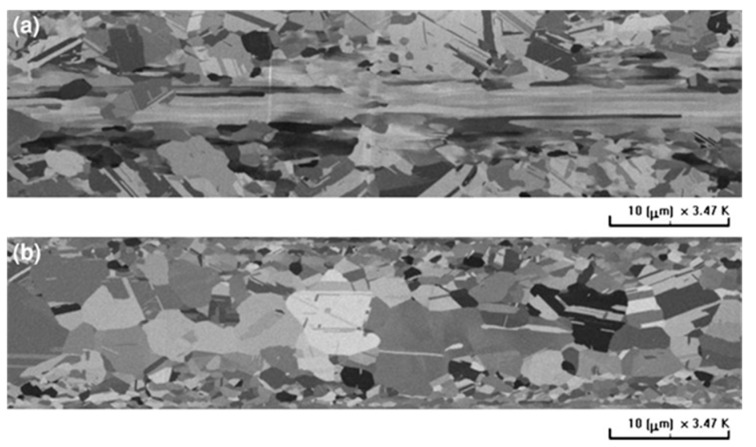
Grain structure in the longitudinal cross-sections of the original bonding wires: (**a**) annealed-twinned Ag-8Au-3Pd wire; (**b**) traditional Ag-8Au-3Pd wire [147]. Copyright, 2012, Springer Nature.

**Figure 13 micromachines-14-00432-f013:**
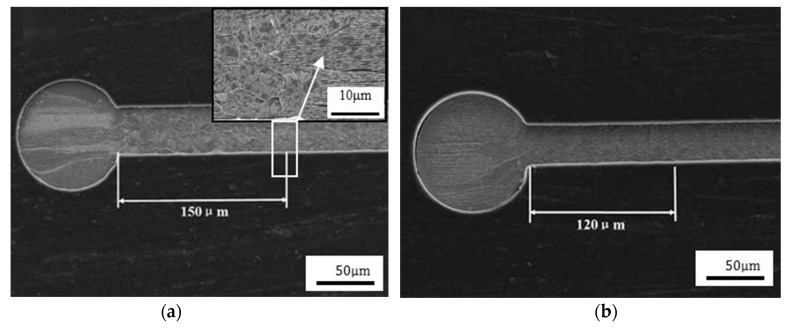
HAZ length of Ag alloy wire (**a**) Ag-1.7Pd wire; (**b**) Ag-1.7Pd-0.75Zn wire [155]. Reproduced with permission from Jun Cao et al.; published by IEEE, 2015.

**Figure 14 micromachines-14-00432-f014:**
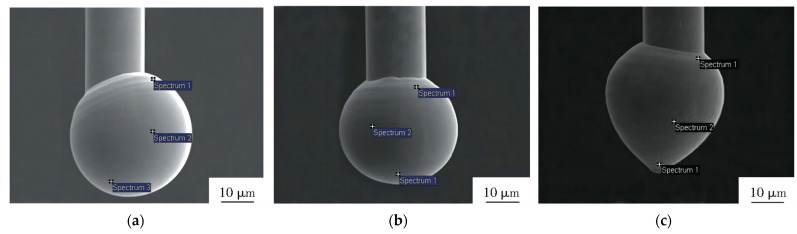
Morphology of FAB for Au-coated Ag bonding wires with (**a**) 48.7 nm, (**b**) 108 nm, and (**c**) 174.7 nm Au coating thickness [156]. Reproduced with permission from Ref. [156]; published by Materials Science and Technology, 2018.

**Figure 15 micromachines-14-00432-f015:**
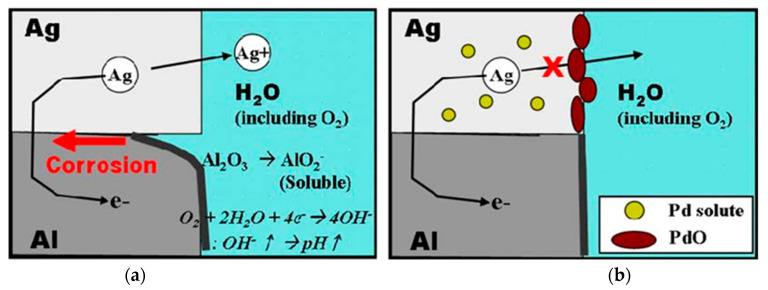
Mitigation mechanism of corrosion: (**a**) bare Ag wire; (**b**) Ag-3Pd alloy wire (Effect on Pd alloying-Ag migration model) [163]. Copyright, 2012, Springer Nature.

**Table 1 micromachines-14-00432-t001:** Advantages of Au alloy wires versus traditional Au wire [32,33].

Alloy Wire Type	Advantages
Au-Pd system	Higher strength, better free air ball (FAB) morphology, lower IMCs growth
Au-Cu system	Higher strength, lower environmental requirements, lower IMCs growth
Au-In system	Better wire loop, lower break rate, higher bonding strength
Au-Cu-Ca system	Higher strength, better wire thinning effect, lower IMCs growth

**Table 2 micromachines-14-00432-t002:** Reliability test results: deteriorated ratios after tests [115]. Copyright, 2006, IEEE.

Test	Duration	Failures by Wire Type		
		Pd-Coated Cu wire	Au Wire	Cu Wire
Temperature Cycle (−65 to −150 °C)	1000 cycles	0%	0%	91%
Temperature humidity bias (85 °C, 85%RH, 10V)	1000 h	0%	0%	3%
Pressure cooker test (PCT) (121 °C, 100%RH, 2 atom)	125 h	0%	0%	95%
Solder reflow (260 °C)	3 times	0%	0%	0%

**Table 3 micromachines-14-00432-t003:** Basic properties of Au, Cu and Ag [17,173,174,175].

Properties	Au	Cu	Ag
Electrical Conductivity (% IACS)	73.4	103.1	108.4
Resistivity (×10^−9^ Ω·m)	23.5	16.7	14.7
Thermal Conductivity (W/m·K)	317.9	398.0	428.0
Thermal Expansion Coefficient (μm/m·K)	14.2	16.7	19.0
Tensile Strength (MPa)	103.0	209.0	125.0
Yield Strength (MPa)	30.0–40.0	33.3	35.0
Elastic Modulus (GPa)	78.0	128.0	71.0
Brinell Hardness (HB)	18.0	37.0	25.0
Metal Activity	Cu > Ag > Au		

**Table 4 micromachines-14-00432-t004:** Challenges and solutions for bonding wires.

Type	Technical Challenges	Solution
Au	Poor loop formability and easy to collapse during wire bonding.	Add alloying elements such as Cu and Pd; use Cu bonding wire to replace it.
	Easy to generate Kirkendall voids under long-term high temperature storage life (HTSL) tests.	Add trace alloying elements to slow down the diffusion rate of Au atoms.
Cu	Easy to be oxidized and corroded.	Chose alloying and plating treatment; use inert shielding gas.
	High hardness, causing defects such as pad peeling and cratering.	Use softer copper wire; increase pad thickness; set dummy microvias beneath pad metallization to stabilize and strengthen pad structure.
	Low hardness and strength of the HAZ, resulting in wire breakage near the FAB.	Adjust the EFO parameters.
	Short circuit and tail defects.	Use Pd-coated Cu wire instead of bare Cu wire.
	Cu-Al IMC corrosion, causing microcracks.	Use low halogen of molding compound and solder resist in the pad.
	Al(OH)_3_ forms during bonding process, galvanic and pitting corrosion occur with the presence of chloride halides in sodium chloride solutions.	Control humidity and the temperature of production workshop [176].
Ag	Easy to be oxidized and vulcanized.	Coat Au, Pd, etc. on bare Ag wire; add alloying elements to make Ag alloy wire.
	Ag^+^ migration.	Add Pd to inhibit Ag^+^ migration.
	Low PCT reliability of bare Ag wire.	Increase Pd content reasonably.
	Complex bonding process with narrow process window.	Optimize the bonding process; add alloying elements such as Au and Pd.
	Short tailing during wire bonding.	Reduce the second bonding force by 10%.
	Poor FAB size repeatability and concentricity of Ag alloy wire.	Adopt lower FAB flow rates and EFO currents [72].
	FAB surface deformation and defects.	Increase Pd content.

## Data Availability

Not applicable.

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
