# Peer review of "Research Progress on Bonding Wire for Microelectronic Packaging"

_micromachines, 2023, doi:10.3390/mi14020432_

Round 1
Reviewer 1 Report
1. Grammatical error on the paper title.
2. Most of the literature in each subsection is good with the critic's element.
3. The references are almost updated and sufficient for the review paper.
4. There is a lot of finite element, CFD, and Fluid-Structure Interaction studies on the microelectronics wirebond process. Please add to the literature
5. The gap and novelty of the wire bonding technology are unclear. Please revise the summary part.

Reviewer 2 Report
#1. The introduction about the background is too long and must be streamlined.
#2. The language needs to be polished.
#3 The manuscript reviews the influence of different material types on bonding quality, however, the content of the manuscript in the present form is not rich enough, as there are already some similar articles over the years, the authors are suggested to refine more novel features.
#4 The authors are also suggested to refine the trend of wire bonding beyond the material aspect, such as the application of artificial intelligence technology in wire bonding (Machine learning enables accurate wire loop profile prediction for advanced microelectronics packaging, Journal of Manufacturing Processes, 2022, 84, 394-402)
#5. The content needs to keep up with the times, there are few references in recent years, it is recommended to add some literature published in the last three years, for example 10.1109/ACCESS.2020.3037338, et al.
Reviewer 3 Report
It is an interesting review but I would like to make certain suggestions to improve this work:
1) Why do the authors use the word "Bonding Wire" (In the title as well as elsewhere in the text) while there exists a standard term like "Wire Bonding"?
2) at line 42, the text sentence is incomplete. ".... accounts for 1/4 -1/3" 1/4 - 1/3 of what? please correct this.
3) Various abbreviations were used without definition. Such as, "IMP formation" at line 91, "HAST" at line 332, "UHAST" at line 349.
4) For all of the images used from other papers, you need to get the permission from the publisher to use these pictures. It seems like the authors did not get such permissions.
5) Quality of certain images should be improved, such as Figure 2, Figure 4a, Figure 7a.
6) At line 231, the manuscript talks about "Ce or Ti" what is "Ce"? please correct this.
Round 2
Reviewer 2 Report
The technical aspects of the article are fine, but the language is recommended to polish